:◌: PLOS | ONE

# Differences in species diversity, biomass, and soil properties of five types of alpine grasslands in the Northern Tibetan Plateau

**Beibei Zhang**[1,2], **Hui Zhang**[1]*, **Qi Jing**[1], **Yuexuan Wu**[1], **Shuqin Ma**[3]

**1** Shaanxi Key Laboratory of Disaster Monitoring and Mechanism Simulating, College of Geography and Environment, Baoji University of Arts and Sciences, Baoji, China, **2** Center for Ecosystem Science and Society, Northern Arizona University, Flagstaff, AZ, United States of America, **3** College of Tourism, Henan Normal University, Xinxiang, China

* huizai207@126.com

**Data Availability Statement:** All relevant data are within the paper and its Supporting Information files.

## Abstract

Approximately 94% of the land area of the Northern Tibetan Plateau is covered by grasslands, which comprise one of five key livestock producing regions in China. In contrast to most other regions worldwide, these alpine grasslands are much more sensitive to global climate change, thus they are under intense study. The differences in species diversity, plant biomass, and soil properties of five representative's alpine grassland types in the Northern Tibetan Plateau were investigated in this research. The results revealed that 11 community types were identified according to the importance of dominant species and constructive species. There were significant differences in the Margalef index (H), Simpson diversity index (D), Shannon-wiener diversity index (H'), and Pielou evenness index (J) indices between these five alpine grasslands. Further, the above-ground biomass (AGB), below-ground biomass (BGB), total biomass (TB), root:shoot (R/S) ratio, and coverage showed significant differences in 5 alpine grasslands. There were also considerable variations in the pH, total nitrogen concentration (TN), total phosphorus concentration (TP), soil organic carbon (SOC) and C-to-N ratio (C:N) among the five alpine grasslands. The highest value of biomass and soil characteristics was always in the alpine steppe (AS), or AM, while the lowest of that was in the alpine desert steppe (ADS), or alpine desert (AD). Moreover, there were significant differences in the soil particle size fractions between the five alpine grasslands. In the AM and AS, the dominant soil particle was clay, while in the alpine meadow-steppe (AMS), ADS, and AD it was fine and medium sand. Substantial correlations were found between the biomass and species diversity indices H, D or H' and soil TN, TP, or SOC. Moreover, silt had a significantly positive correlation with soil C:N, BGB, TB, and R/S, while medium sand and coarse sand was significant negatively correlated. With regard to these grassland types, it is proposed that the AM or AS may be an actively changing grassland types in the Northern Tibetan Plateau.

**Funding:** This work was supported by the National Natural Science Foundation of China (NSFC, No. 41601016), Special Support Program for High-Level Personnel Recruitment (Youth Top-Talent) of Shaanxi province, National Natural Science Foundation of Shaanxi province (No. 2017JQ4023), The Key Project of Baoji University of Arts and Sciences (No. ZK2018012).

**Competing interests:** The authors have declared that no competing interests exist.

## Introduction

The Tibetan Plateau, known as "the Third Pole of the Earth", comprises ~ 2.3 million $km^2$ of alpine grasslands with a mean elevation of more than 4000 m. These very important pasturelands are the highest and largest alpine grassland regions in the world [1,2] and are also a differentiation center for new plant species [3]. In this region, terrestrial ecosystems and the ambient atmosphere interact and contribute to the establishment of diverse biomes and unique vegetation patterns [4].

These alpine grasslands are not only the most important and largest ecosystem in this area, occupying approximately 94%, but are also the key resources that support the subsistence of the local population [5]. The main types of natural grasslands in this area are alpine meadow-steppe (AMS), alpine meadow (AM), alpine steppe (AS), alpine desert steppe (ADS), alpine desert (AD) [6,7], while each has its own dominant species, with most plants being perennial herbs [8].

Species diversity is one of the most important community attributes that influences stability, productivity, and migration [9]. Variations in species diversity might be linked to several factors such as locations or grassland types [10]. Species diversity primarily includes species richness and evenness, which include many indices such as Patrick species richness index (R), Margalef diversity index (H), Simpson diversity index (D), and Shannon-wiener diversity index (H') and so on, allof which can reflect the characteristics of plant communities [11]. Thus, an improved elucidation of the correlations between species diversity and plant growth might assist with understanding the the overall functionality of grassland ecosystems [12].

Aboveground biomass (AGB) and belowground biomass (BGB) are the most critical elements for plant growth, as the major contributors to soil organic matter, which impacts greenhouse gas emissions and carbon (C) cycles in terrestrial ecosystems; thus, biomass has a particular functions in the global climate change and carbon sequestration [13]. Biomass values are also a crucial prerequisite for the estimation of C stocks and pools [14]. For the biomass researches, the allocation of AGB and BGB are core parameters in plant ecology [1]. Moreover, they are also was the results of the long-term adaptation of plants to environmental conditions, as well as the overall functions of the ecosystems and biogeochemical cycles [15].

Soil nitrogen (N) and phosphorus (P) are vital minerals that limit primary production in terrestrial ecosystems [16], and important determinants of species richness, evenness, and community composition. Furthermore, soil particle size fractions constitute another critical soil attribute that might influence soil properties, such as soil water retention, soil thermal conductivity, soil sorption properties, soil nitrification, denitrification, and many other soil properties [17].

Previous studies have set their focus mainly on the differences of species diversity, biomass, and soil properties under alpine meadow ecosystems [11,18]. Thus, the objective of this study was to learn the differences between five alpine grassland types on species diversity, biomass, and soil property parameters in the Nnorthern Tibetan Plateau.

## Material and methods

### Study site

The grassland alpine meadow-steppe (AMS), alpine meadow (AM) and alpine steppe (AS) samples for this experiment were selected in the Naqu Prefecture, which is located between 29˚55' and 36˚30'N, and from 83˚55' to 95˚5'E, and covers 394,632 $km^2$ in Northeastern Tibet. The average altitude is ~4500 m with the landscapes being nestled between the Kunlun, Tonggula, and Nieqintonggula mountain ranges. The mean annual temperature ranges from -0.9˚C to—3.3˚C, and the annual relative humidity is from 48%-51% [19]. The mean annual precipitation is ~380 mm, which occurs mainly during the short cool summer, with ~2580 h of sunlight annually.

The alpine desert steppe (ADS) and alpine desert (AD) samples were selected in Ngari Prefecture, which is located 30˚ and 35˚50' N and from 78˚3' to 86˚ E, spanning 350,000 km$^2$. This area has an average altitude 4500 m, while central and eastern Ngari comprise the western portion of the Qiangtang Plateau that is characterized by an extremely continental plateau climate. The annual mean temperature was quite low with 3˚C in the south, -0.1˚C in the central region, and as low as—10˚C in the north. The annual precipitation is only ~180 mm [20].

The study sites, geographical locations, and sample data are shown in Fig 1 and Table 1; these five types of alpine grasslands were grazed and didn't have any other management. The sampling duration was from August 4[th] to 17[th] in 2016. No specific permissions were required for these locations and the field studies did not involve endangered or protected species.

## Plant and soil materials

For the AM, AS, and ADS, five sample plots (50m×50m) were selected, while three AMS and AD sample sites were studied. Fore each sample plot, three different quadrats (1m×1m) were investigated for species diversity, which included the species composition, abundance, and coverage. The community differences, species diversity and importance value data were showed as the supporting information (S1 Table). The above-ground biomass (AGB) of each quadrat was quantified by clipping at the soil-surface level, while the below-ground biomass (BGB) was also measured. The plant samples were weighed following oven-drying at 80˚C for 48 h [11]. The top soil layers were sampled at depths of from 0–15 cm. Once the soil water content was measured, the soil samples were carefully sifted through a 60-mesh sieve, and loaded into bags to measure the pH, N, P, soil organic C and soil particle size.

## Methods

### Importance value (IV)

The calculation formula could be expressed as follows:

$$IV = \frac{D + C + F}{3} \tag{1}$$

Where D is the relative density, C is th relative cover rate, and F is the relative frequency.

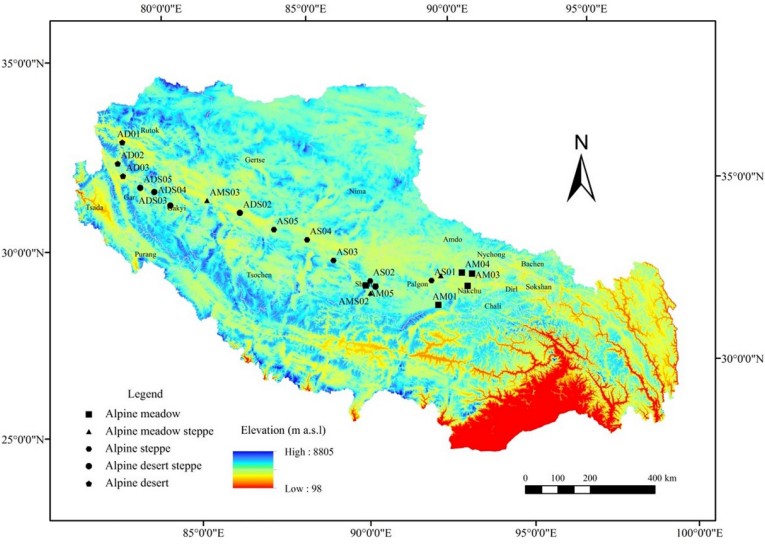

**Fig 1. Research areas and sampling sites.**

**Table 1. Sample geographical locations and additional data for the five alpine grasslands.**

| Grassland type | Site | Latitude N | Longitude E | Elevation |
|---|---|---|---|---|
| **Alpine Meadow (AM)** | Nam Co | 91.112 | 30.750 | 4812 |
| | Naqu | 91.980 | 31.377 | 4594 |
| | Nima | 92.070 | 31.729 | 4670 |
| | Anduo | 91.730 | 31.716 | 4655 |
| | Shenzha | 88.699 | 30.957 | 4654 |
| **Alpine Steppe (AS)** | Bange | 90.777 | 31.389 | 4619 |
| | Shenzha | 88.700 | 31.124 | 4735 |
| | Nima | 87.483 | 31.505 | 4648 |
| | Nima | 86.503 | 31.932 | 4718 |
| | Gaize | 85.356 | 32.032 | 4785 |
| **Alpine Meadow-Steppe (AMS)** | Bange | 91.058 | 31.552 | 4544 |
| | Shenzha | 88.702 | 30.957 | 4664 |
| | Gaize | 82.978 | 32.429 | 4421 |
| **Alpine Desert Steppe (ADS)** | Shenzha | 88.712 | 30.960 | 4733 |
| | Nima | 84.133 | 32.291 | 4438 |
| | Gaize | 81.826 | 32.071 | 4606 |
| | Gaize | 81.206 | 32.327 | 4543 |
| | Ge'gyai | 80.715 | 32.347 | 4628 |
| **Alpine Desert (AD)** | Rutog | 79.755 | 33.432 | 4266 |
| | Rutog | 79.784 | 32.838 | 4443 |
| | Rutog | 80.061 | 32.544 | 4400 |

## Species diversity

$$\text{Patrick species richness index (R) :} \quad R = S \tag{2}$$

$$\text{Margalef diversity index (H) :} \quad H = \frac{(S-1)}{LnN} \tag{3}$$

$$\text{Simpson diversity index (D) :} \quad D = 1 - \sum_{i=1}^{S} P_i^2 \tag{4}$$

$$\text{Shannon}-\text{Wiener diversity index (H') :} \quad H' = -\sum_{i=1}^{S} P_i LnP_i \tag{5}$$

$$\text{Pielous evenness index(J) :} \quad J = \frac{(H)}{LnS} \tag{6}$$

Where S is the species number of the sampling quadrats, Pi represents the relative importance value of species i, N is the total number of sampling quadrats [21,22].

## Soil analysis

The soil samples were ground using a Spex Sample Prep 8000D ball mill (Metuchen, NJ, USA) to a fine power. The soil pH was measured using a PH 3000 (Steps, Germany). The soil organic carbon (SOC) was determined using wet oxidation by $K_2Cr_2O_7\text{-}H_2SO_4$ [23]. The soil samples were digested using a sulfuric acid-hydrogen peroxide digestion procedure, after which a FLOWSYS III system (Flowsys, Systea, Itally) was employed to measure the total N and P. The soil particle size fractions were determined using a laser diffraction instrument (Malvern

Mastersizer 2000 particle size analyzer, Worcs, UK), which is the classification used by the U. S. Department of Agriculture (USDA). This is: clay (<2 μm), silt (2–50 μm), very find sand (50–100 μm), fine sand (100–250 μm), medium sand (250–500 μm) and coarse sand (500–2000 μm) [24].

## Statistical analysis of data

All collected data were subjected to one-way ANOVA in SPSS analysis (SPSS software version 25.0, Chicago, Illinois, USA). The differences between means were compared by Tukey's HSD test at P < 0.05. Redundancy analysis (RDA) was based on a covariance matrix, and was conducted to evaluate how the species diversity and soil property parameters were interrelated between different alpine grassland types using the package CANOCO package, version 5.0 (Microcomputer Power, Inc., Ithaca, NY). Correlations between parameters were determined using the Pearson's simple correlation test function of SPSS.

## Results

### Community differences and species diversity between five alpine grassland types

According to the importance of dominant species and constructive species in five alpine grasslands, 11 community types were identified and the main species importance values (IV) are shown in supporting information (S1 Table). For three communities in the alpine meadow (AM), the main species were *Carex*, *Poa pratensis* and *Kobresia humilis*, where the IV were 0.257, 0.217, and 0.174, respectively. For three communities in the alpine desert (AD), the main species were *Suaeda*, *Stipa*, and *Artemisia wellbyi*.

The species diversity characteristics are revealed in Table 2. There were significant differences in the Margalef index (H), Simpson diversity index (D), Shannon-wiener diversity index (H'), and Pielou's evenness index (J) indices between 11 communities, which also had significant differences between five alpine grasslands. The AM communities, they had higher R, H, and D than any other alpine grasslands while in the AMS, they had higher H' and J than any other grasslands.

Significant correlations were observed between H and D, H' (r = 0.715, 0.531, respectively; P<0.01; data not shown), while the negative correlations between H and J were not significant. We learned that D was positively correlated with H' (r = 0.641**, P<0.01), while it was not significant with J, and H' was significantly correlated with J (r = 0.669**, P<0.01).

### Coverage and biomass differences between five alpine grasslands

There were significant differences between the aboveground biomass (AGB), belowground biomass (BGB), total biomass (TB), root:shoot (R/S) ratio, and coverage between the 11 communities (Table 3). The BGB, TB, coverage, and R/S were the highest in the AM1 community, while there were lowest in the AD3 community. The AGB varied minimally from the BGB and TB, while the highest AGB was in the AM3 community and lowest in the AD2 community.

The AGB, BGB, TB, R/S, and coverage of the five different alpine grasslands were also shown in Table 3. There was a significant difference in the mean coverage between these five types alpine grasslands (F = 36.067, P<0.01), where in the AM, the coverage was 40.330, which was much higher than any of the other four alpine grasslands, while the lowest coverage was in the AD, at only 16.260. The R/S differed significantly between the five alpine grasslands (F = 49.423, P<0.01), while the highest was in the AM, and the lowest was in the AD.

**Table 2. Species diversity indices for the five alpine grasslands.**

| Grassland type | Community | R | H | D | H' | J |
|---|---|---|---|---|---|---|
| AM | AM1 | 6 | 0.602±0.006c | 0.968±0.003a | 1.563±0.010d | 0.873±0.009d |
| | AM2 | 7 | 0.693±0.003a | 0.863±0.002bc | 1.643±0.012c | 0.845±0.003e |
| | AM3 | 6 | 0.630±0.003b | 0.871±0.002b | 1.668±0.009c | 0.931±0.008c |
| AS | AS1 | 6 | 0.629±0.004b | 0.764±0.001d | 1.359±0.008f | 0.759±0.004f |
| | AS2 | 5 | 0.592±0.003cd | 0.864±0.003bc | 1.465±0.009e | 0.910±0.006c |
| AMS | AMS1 | 4 | 0.405±0.005e | 0.639±0.009e | 2.037±0.012b | 1.469±0.005b |
| | AMS2 | 4 | 0.381±0.002f | 0.726±0.003de | 2.141±0.013a | 1.545±0.007a |
| ADS | ADS | 3 | 0.324±0.006g | 0.624±0.004e | 0.817±0.009h | 0.654±0.002g |
| AD | AD1 | 3 | 0.202±0.007hi | 0.53±0.006i | 0.927±0.006g | 0.843±0.001e |
| | AD2 | 2 | 0.169±0.002i | 0.547±0.002hi | 0.367±0.006j | 0.53±0.003h |
| | AD3 | 2 | 0.231±0.003h | 0.597±0.004h | 0.511±0.004i | 0.738±0.004f |
| AM | | 7 | 0.642±0.014a | 0.893±0.011a | 1.624±0.017b | 0.883±0.013b |
| AS | | 6 | 0.610±0.009a | 0.811±0.022b | 1.414±0.025c | 0.835±0.034b |
| AMS | | 4 | 0.394±0.007b | 0.684±0.020c | 2.090±0.023a | 1.509±0.018a |
| ADS | | 3 | 0.321±0.006b | 0.621±0.006c | 0.889±0.006d | 0.751±0.006c |
| AD | | 2 | 0.200±0.062c | 0.553±0.022d | 0.603±0.084e | 0.703±0.046d |
| Analysis of variance | F value of 11 different communities | | 847.337** | 316.526** | 9853.516** | 3997.036** |
| | F value of 5 grasslands | | 11.710** | 24.713** | 118.696** | 131.916** |

Values presented in the first section of table are mean±standard errors. Last section of the table means the F value.

** P<0.01. Margalef diversity index (H), Simpson diversity index (D), Shannon-Wiener diversity index (H'), Pielou's evenness index (J).

**Table 3. Differences in aboveground biomass (AGB), belowground biomass (BGB), total biomass (TB) and root:shoot (R/S) ratio in the five alpine grasslands.**

| Grassland type | AGB (g·m⁻²) | BGB (g·m⁻²) | TB (g·m⁻²) | R/S | Coverage (%) |
|---|---|---|---|---|---|
| AM1 | 18.100±0.127d | 331.753±2.630a | 349.853±2.534a | 13.838±0.259a | 44.000±0.577a |
| AM2 | 24.917±1.144b | 109.477±5.140b | 134.393±6.279b | 4.393±0.022c | 45.000±0.577a |
| AM3 | 29.613±0.875a | 116.050±5.039b | 145.661±5.892b | 5.625±0.040b | 32.000±0.577c |
| AS1 | 18.830±0.278d | 88.610±0.527c | 107.440±0.613c | 4.708±0.074c | 23.400±2.862ef |
| AS2 | 13.424±0.216e | 58.602±1.486d | 72.026±1.674d | 4.194±0.079c | 26.000±3.055de |
| AMS1 | 12.880±0.168e | 53.918±4.319d | 67.114±4.487de | 3.278±0.252d | 28.000±0.577d |
| AMS2 | 21.654±1.964c | 40.633±4.746e | 62.970±6.704de | 1.810±0.055e | 20.000±0.559f |
| ADS | 12.133±0.912e | 43.644±1.321e | 55.777±7.358e | 3.322±0.312d | 18.530±0.291fg |
| AD1 | 9.960±0.220f | 32.077±1.185f | 42.037±1.405f | 3.219±0.047d | 15.100±0.586g |
| AD2 | 8.410±0.186f | 34.247±1.039f | 42.657±1.223f | 1.271±0.037f | 18.530±0.291fg |
| AD3 | 10.540±0.438f | 9.378±0.801g | 27.761±1.239g | 1.627±0.009e | 15.130±0.593g |
| F value | 64.410** | 824.838** | 611.603** | 86.2828** | 61.225** |
| AM | 24.209±3.402a | 185.760±25.283a | 209.969±28.490a | 7.952±0.481a | 40.330±2.108a |
| AS | 16.127±1.793b | 73.606±9.217b | 89.733±10.966b | 4.451±0.198b | 24.700±1.960b |
| AMS | 17.767±1.347b | 47.274±8.196c | 65.042±9.397bc | 2.544±0.038cd | 24.000±1.826b |
| ADS | 12.133±0.912b | 43.644±6.513c | 55.777±7.358bc | 3.322±0.312c | 18.530±0.291bc |
| AD | 12.251±0.654b | 25.234±2.520c | 37.485±3.010c | 2.039±0.156d | 16.260±0.624c |
| F Value | 5.921** | 20.515** | 18.319** | 49.423* | 36.067** |

Values presented in the first section of table are mean±standard errors. The last section of table means the F value.

* P<0.05

** P<0.01.

Moreover, the BGB was also showed significant differences between them (F = 20.515, P<0.01), while in the AM it was 185.760 g·m$^{-2}$, and in the AD it was only 25.234 g·m$^{-2}$.

The AGB was significantly correlated with BGB, TB, R/S, and coverage (data not shown, r = 0.888, 0.908, 0.422, 0.582, respectively; P<0.01). The BGB had a positive correlation with the TB, R/S, and coverage (r = 0.999, 0.901, 0.744, respectively; P<0.01). TB was significantly correlated with R/S and coverage (r = 0.720, 0.788, respectively; P<0.01). Moreover, there was a significant correlation between the R/S and coverage (r = 0.595, P<0.01).

## Differences in soil properties between five types of alpine grasslands

There were significant differences between the 11 communities in terms of soil water content, pH, total nitrogen concentration (TN), total phosphorus concentration (TP), soil organic carbon (SOC), and C-to-N ratio (C:N) (Table 4), and which were also significantly different in five types of alpine grasslands. The lowest soil water content was in the AD at only 2.612%. For the pH results, all of the soil samples were alkaline with AMS having the highest pH. The highest TN was in the AM (1.819 g·kg$^{-1}$), while the lowest was in the AD (0.086 g·kg$^{-1}$). The TP of the AS was the highest, while the ADS had the lowest. The differences in SOC were highest in the five alpine grasslands (F = 262.484, P<0.01), where the SOC of the AM was highest and the lowest in the AD. Moreover, the difference in C:N was the lowest (F = 4.058, P<0.05).

As can be seen in Fig 2, there were considerable differences in the soil particle size fractions between the five alpine grassland types. In the AM and AS, the dominant soil particle was clay and the value was lower around 1%; however, in the AMS, ADS, and AD, the dominant soil particles were fine and medium sand. Overall, sand, silt and clay fractions changed clearly changed and were significantly different across the various grassland types.

**Table 4. Differences in soil pH, total nitrogen concentration (TN), total phosphorus concentration (TP), soil organic carbon (SOC) and C-to-N ratio (C:N) in the five alpine grasslands.**

| Grassland type | pH | TN (g·kg$^{-1}$) | TP (g·kg$^{-1}$) | SOC (g·kg$^{-1}$) | C:N |
|---|---|---|---|---|---|
| AM1 | 7.680±0.012f | 1.195±0.008c | 0.787±0.005d | 27.090±0.068c | 22.671±0.191c |
| AM2 | 7.733±0.009e | 2.427±0.049a | 0.946±0.008b | 36.184±0.103aa | 14.924±0.341e |
| AM3 | 8.033±0.009d | 1.834±0.027b | 0.926±0.003bc | 35.527±0.259b | 19.385±0.417d |
| AS1 | 7.450±0.006g | 0.736±0.008e | 1.194±0.037aa | 14.905±0.086e | 20.266±0.355d |
| AS2 | 7.643±0.009f | 0.974±0.013d | 0.911±0.001c | 15.394±0.072d | 15.815±0.246e |
| AMS1 | 9.563±0.033a | 0.271±0.008g | 0.579±0.001e | 6.936±0.113g | 25.689±0.896b |
| AMS2 | 8.943±0.003b | 0.612±0.008f | 0.601±0.002e | 7.917±0.094f | 12.947±0.314f |
| ADS | 8.970±0.007b | 0.170±0.001h | 0.120±0.001i | 4.035±0.051h | 23.750±0.395c |
| AD1 | 8.096±0.003c | 0.055±0.002i | 0.187±0.001h | 1.635±0.041j | 29.615±1.454a |
| AD2 | 8.017±0.017d | 0.052±0.002i | 0.262±0.003g | 1.417±0.046j | 27.151±0.559b |
| AD3 | 8.133±0.017c | 0.151±0.004h | 0.375±0.002f | 2.404±0.039i | 15.897±0.154e |
| F value | 2165.722** | 185.278** | 951.883** | 15533.664** | 83.552** |
| AM | 7.816±0.055d | 1.819±0.179a | 0.886±0.025b | 32.934±1.466a | 18.998±1.134b |
| AS | 7.547±0.043e | 0.855±0.054b | 1.053±0.066a | 15.150±0.120b | 18.043±1.012b |
| AMS | 9.253±0.139a | 0.442±0.077c | 0.590±0.005c | 7.437±0.229c | 19.315±2.881ab |
| ADS | 8.970±0.021b | 0.170±0.001d | 0.120±0.001e | 4.035±0.051d | 23.757±0.392ab |
| AD | 8.082±0.018c | 0.086±0.016e | 0.275±0.027d | 1.818±0.151e | 24.210±2.164a |
| F Value | 98.453** | 43.089** | 109.777* | 262.484** | 4.058* |

Values presented in the first section of table are mean±standard errors. The last section of table means the F value.

\* P<0.05

\*\* P<0.01.

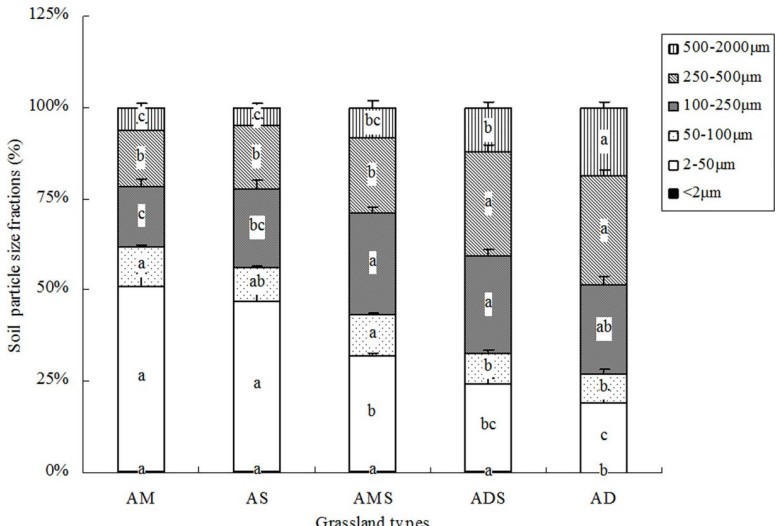

**Fig 2. Soil particle size fraction in the five alpine grasslands.** AM: Alpine meadow, AS: Alpine steppe, AMS: alpine meadow steppe, ADS: alpine desert steppe, AD: alpine desert; error bars were standard errors.

Moreover, the correlations between the soil particle size fractions and soil properties were analyzed (data not shown). The clay was positively correlated with TP (r = 0.355, P<0.01), while it was negatively correlated with pH (r = -0.393, P<0.01). The silt was positively correlated with water content, TN, and SOC (r = 0.301, 0.515, 0.628, respectively; P<0.05), while it was negatively correlated with pH (r = -0.531, P<0.01). There were significantly negative significant correlations between the fine sand and soil water content, TN, and SOC (r = -0.405, -0.518, -0.612, respectively; P<0.01). The medium sand showed a negative correlation with TN and SOC (r = -0.429, -0.525, respectively; P<0.01), while it had a positive correlation with pH (r = 0.517; P<0.01). The coarse sand had a negative correlation only with SOC (r = -0.314; P<0.05), while it showed a positive correlation with pH (r = 0.527; P<0.01).

## Correlations between biomass and species diversity or soil properties in five types of alpine grasslands

The ordination of the plots according to the results of the redundancy analysis (RDA) based on the parameters of species diversity (H, D, H', and J), biomass (AGB, BGB and TB) and soil properties (pH, TN, TP, SOC and C:N) confirmed the correlation between them (Fig 3). The first axis (pseudo-F = 77.76, P = 0.001) and all the axes of the RDA (pseudo-F = 20.8, P = 0.001) were significant. Most parameters of species diversity (H, H', J and D), biomass (AGB,TGB and TB), and soil properties (SWC, TN, TP, and SOC) were related to this axis. In the opposite direction, pH and C:N showed an interrelationship to this axis.

Significant correlations were found between the biomass (AGB, BGB, and TB) and species diversity indices H, D, and H', while the correlation between the biomass and J was not significant (Table 5). The soil TN, TP, and SOC were significantly positively correlated with the biomass, while the soil pH did not have significant correlation with them. Further only C:N had a significant negative correlation with BGB (r = -0.518, P<0.01). For soil particle size fractions, silt exhibited a significant positive correlation with BGB, TB, and R/S (r = 0.368, 0.354, 0.414, respectively; P<0.01) while the medium and coarse sands were significantly negatively correlated with them.

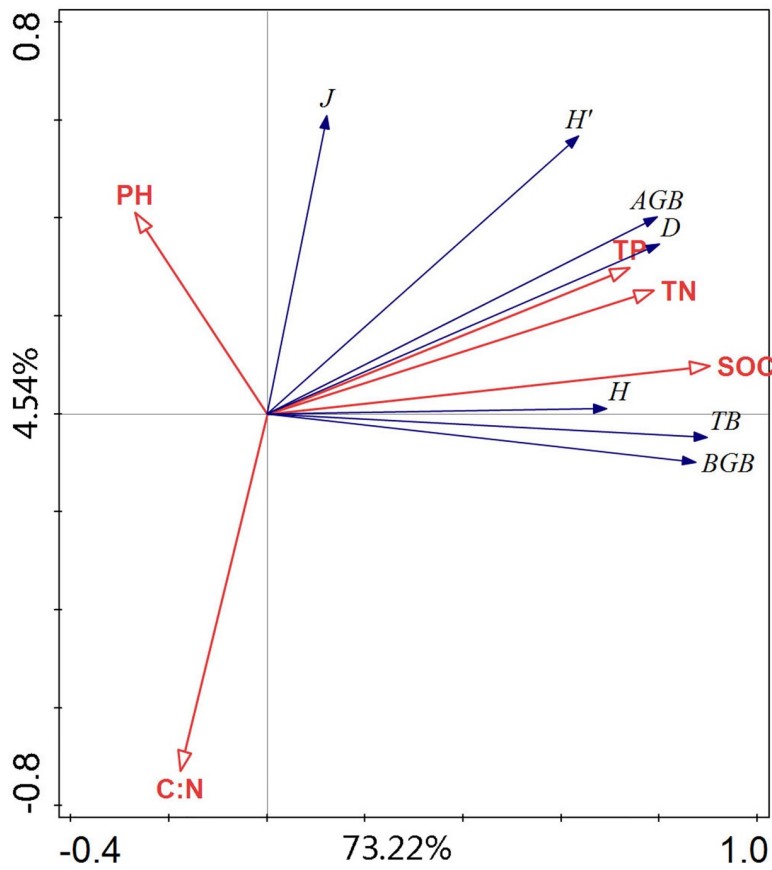

**Fig 3. Redundancy analysis (RDA) of species diversity, biomass and soil properties in the five alpine grassland types.** Margalef diversity index (H), Simpson diversity index (D), Shannon-Wiener diversity index (H'), Pielou's evenness index (J), pH, total nitrogen concentration (TN), total phosphorus concentration (TP), soil organic carbon (SOC), C:N, above-ground biomass (AGB), below-ground biomass (BGB), total biomass (TB).

## Discussion

### Species diversity and biomass

The maintenance of species diversity has emerged as an important topic in grassland management studies, with a special emphasis on elucidating the role of various species in the recovery of grassland structures and processes [25,26]. The differences between the various species combinations at a given diversity level measured the effects of alternative species compositions [27].

Zhao et al [28] revealed that dominant species of alpine grasslands on the northern Tibetan Plateau have differential strategies in foliar nutrient resorption and growth to adapt to the limitation nutrient and water in desert steppes. Some other findings also suggested that the biodiversity-ecosystem function relationship can be regulated by species composition and interspecific interactions [29]. In our research, the important value (IV) is investigated and calculated in sampling sites. These IV is used to classify the dominated and constructive species to 11 communities and each species has its own function in contributing ecosystem balance [19,30].

For species diversity we employed a number of indices for its investigation. Among these indices, R was a basic measurement used whenever possible for a direct diversity expression

**Table 5. Correlation between species diversity, biomass and soil properties in the five alpine grasslands.**

| Index | AGB | BGB | TB | R/S |
|---|---|---|---|---|
| H | 0.596** | 0.523** | 0.551** | 0.318 |
| D | 0.777** | 0.618** | 0.658** | 0.382* |
| H' | 0.643** | 0.369* | 0.407** | 0.191 |
| J | 0.203 | -0.036 | -0.018 | -0.048 |
| pH | -0.201 | -0.331 | -0.334 | -0.209 |
| TN | 0.832** | 0.539** | 0.586** | 0.227 |
| TP | 0.686** | 0.469** | 0.506** | 0.224 |
| SOC | 0.838** | 0.693** | 0.734** | 0.380* |
| C:N | -0.518** | -0.022 | -0.063 | 0.170 |
| Clay (<2μm) | -0.072 | -0.182 | -0.084 | 0.044 |
| Silt (2–50μm) | 0.196 | 0.368** | 0.354** | 0.414** |
| Very fine sand (50–100μm) | 0.204 | 0.239 | 0.238 | 0.212 |
| Fine sand (100–250μm) | -0.084 | -0.241 | -0.268* | -0.348* |
| Medium sand (250–500μm) | -0.227 | -0.343** | -0.334** | -0.356** |
| Coarse sand (500–2000μm) | -0.200 | -0.314* | -0.306* | -0.286* |

* $P<0.05$

** $P<0.01$.

[31]. The Margalef index served as a concept behind the species-area curve [32], whereas the Simpson index (D) mostly expressed the dominance or relative concentration of the importance values for the first, or first few species. The Shannon-Wiener index (H') expressed the relative evenness or equitability of the importance values through the entire sequence [33,34].

All of these indices reflected the species diversity in different areas. For our study, the highest R, H, and D was in the AM, and the correlations between H, D, and H' were also significant (Table 3). When the evenness and the richness (abundance) were generally higher, the diversity index of the community would be higher; thus, the AM community had the highest diversity.

It has been widely reported that species diversity could affect ecosystem functionality [35]. Some research has suggested that changes in plant species diversity might affect several ecosystem processes, such as biomass production [36,37]. Diversity losses in plant communities could limit plant recruitment and decrease biomass production in plants, which would impact ecosystem functions [38]. Furthermore, the positive impact of species diversity on biomass production has been explained by the complementarity of resource use between plant species, or their functional groups [39,40].

For our research, significant correlations were observed between biomass and species diversity indices H, D, and H' (Table 5). Moreover, previous studies revealed that during changes in species diversity, biomass production, and ecosystem functioning, land use change was highlighted as one of the most immediate causes, with which our results agreed. Besides that, "inertness" always describes as the alpine grassland ecosystems because its ecosystem energy flow and material circulation rate are slower than other ecosystems [41], leading to slow renewal rates of alpine grassland ecosystems under rapid transformation by human activities (grazing). Such as *Kobresia* as a dominant species in these alpine grassland types relies on asexual reproduction and its annual regeneration rate is very slow [42]. In these sites the low decomposition rate of organic matter results in a large amount of undecomposed organic matter accumulated on the surface [43]. Our study also confirmed the results that in AM the SOC concentration was higher than 30 g·kg$^{-1}$ and the correlation between SOC and biodiversity is also significantly positive (Table 4 and Fig 3).

## Soil property and biomass

Most arid alpine grasslands mainly distributed in high altitude regions have been degraded by grazing, and aridity stress [44]. The cold climate in these areas is responsible for soil temperature, soil moisture and soil properties which directly regulate plant growth [45]. Recently, as a result of increase of greenhouse gases emissions, global temperature has been rapidly increasing especially in Northern Tibetan Plateau, 0.2°C per decade over the past half century [46], combined with drought because of increased evapotranspiration and results in decreased biomass [47]. Our result is similar to the former results that the biomass value is decreased compared to the last decades.

In our study, soil properties, including TN, TP, and SOC were different in the five grassland types, which were significantly positively correlated with the biomass (Tables 4 and 5). As is known, soil is a dynamic, living, natural body, and a key factor in the sustainability of terrestrial ecosystems. Its properties have a significant influence on the productivity of ecosystems [48]. However, variability in soil properties, a rule rather than an exception, necessitated site-specific management for optimizing the efficient use of inputs [49].

This variation influences soil functions, such as nutrient mobility and their redistribution and supply to plants, as well as shoots and roots growth [50]. Therefore, soil properties (e.g., soil organic carbon, nitrogen, phosphorus and pH) were critical factors that affected shoots and roots growth, which altered biomass production [51] that is always dependent on the complex interactions between spatially variable physical and chemical properties of soil.

Soil particle size fractions were one of the most important physical attributes of soil properties [52]. With the process of the sandy desertification in grassland, the clay fraction decreased. In our study, the clay fraction was extremely lower under drought sites (Fig 2) which was similar to the results of Su et al [53] that under extremely desertified condition the clay fraction was only around 1%. The silt fraction markedly decreased, while the medium and fine sand fractions increased significantly in the AD and ADS grasslands. This suggested that silt, and very fine sand were selectively removed, which caused progressive coarsening in the desertification process, and initiated changes in the growth of plants.

Moreover, the clay and silt were positively correlated with soil water content, TN, TP, and SOC, while the sand was observed to be negatively correlated. Several studies have reported higher biomass in smaller size fractions, such as clay or silt [54,55]. Our research also identified this, where the silt had significantly positive correlation with BGB, TB, and R/S, while medium and coarse sands were significantly negatively correlated with them (Table 5).

## Conclusions

Alpine grasslands are fragile ecosystems, such as comprise the major types of pasturelands in the Northern Tibetan Plateau. Thus, their growth, geographical situation, conditions, relationships, and differences comprised the main scope of our research. For this study, the differences in the species diversity, biomass, and soil properties of five alpine grassland types in the Northern Tibetan Plateau were investigated in depth. As relates to its grassland types, we suggested that the AM or AS may be an active grassland types in this region. However, we propose that further research, with more seasonal and interannual investigations will be required to evaluate the results.

## Supporting information

**S1 Table. Community, species diversity and importance value in the five alpine grasslands.** (DOC)

## Acknowledgments

The authors are grateful to He Wang and Xuyang Lu for assistance with field-data collection.

## Author Contributions

**Data curation:** Shuqin Ma.

**Formal analysis:** Hui Zhang.

**Funding acquisition:** Beibei Zhang.

**Investigation:** Qi Jing, Yuexuan Wu.

**Writing – original draft:** Beibei Zhang.

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
