## [Decision Letter · Decision Letter 0]

4 Dec 2019

PONE-D-19-25384

Differences in species diversity, biomass, and soil properties of five types of alpine grasslands in the Northern Tibetan Plateau

PLOS ONE

Dear Beibei Zhang

Thank you for submitting your manuscript to PLOS ONE. After careful consideration, we feel that it has merit but does not fully meet PLOS ONE’s publication criteria as it currently stands. Therefore, we invite you to submit a revised version of the manuscript that addresses the points raised during the review process.

DR.Tuneera Bhadauria

ACADEMIC EDITOR

Deptt of Zoology Feroze Gandhi P.G College 

Raebareli,  Uttar Pradesh India

: We would appreciate receiving your revised manuscript by 25 Dec 2015 .To enhance the reproducibility of your results, we recommend that if applicable you deposit your laboratory protocols in protocols.io, where a protocol can be assigned its own identifier (DOI) such that it can be cited independently in the future. For instructions see: http://journals.plos.org/plosone/s/submission-guidelines#loc-laboratory-protocols

We look forward to receiving your revised manuscript.

Kind regards,

Tunira Bhadauria, Ph.D.

Academic Editor

PLOS ONE

Journal Requirements:

Additional Editor Comments (if provided):

Reviewers' comments:

Reviewer's Responses to Questions

**Comments to the Author**

1. Is the manuscript technically sound, and do the data support the conclusions?

Reviewer #1: Partly

Reviewer #2: Partly

2. Has the statistical analysis been performed appropriately and rigorously? 

Reviewer #1: Yes

Reviewer #2: No

3. Have the authors made all data underlying the findings in their manuscript fully available?

Reviewer #1: No

Reviewer #2: Yes

4. Is the manuscript presented in an intelligible fashion and written in standard English?

Reviewer #1: Yes

Reviewer #2: Yes

5. Review Comments to the Author

Reviewer #1: This paper contains an interesting data set, but an uninteresting story. I have concerns about both plant and soil measurements, as the plant biomass seems low even for very arid systems, and the near absence of clay in the soils seems very unusual. The authors could be more convincing in discussing their methods, thereby providing some confidence in the numbers, and also by comparing their results to other, similar findings in the discussion.

Relating plant and soil characteristics can be strengthened. The correlation matrix among the soil variables should be included (at least as supplemental…). The authors should know that TP, TN and SOC are auto-correlated to the point where only one of the variables, along with C:N ratio, should be used in any regression relationship. Soil moisture, unless averaged over the seasons, is a “snapshot measurement” and tells us almost nothing about plant dynamics.

The value of presenting findings of 11 communities within 5 grasslands is not discussed. If the variances in variables justifies this separation, then OK but this needs to be explicit.

The literature cited on biodiversity-productivity relationships as well as on soil-plant feedbacks is dated. Newer summaries and syntheses are available. Species richness of these sites seems very, very low compared to other cold region/dry sites, and is worthy of comment. Historical legacies (i.e., impacts of vertebrate grazing) need to be presented so readers know of its significance (either large or small….).

There could be a compelling story here by making comparisons to similar, published data on cold region herbaceous systems, but, alone, the work doesn't merit publication in an international journal. Sensitivity responses based upon measurements presented here are tenuous at best.

Specific comments

Abstract: Line 38: The highest value (of what?) was always…

Introduction, line 83: “influences stability” but does not “determine stability”.

Lines 92-96. Actually, carbon sequestration can be independent, at least of AGB.

Line 109: you have no data on change in soil texture so this statement doesn’t support your work.

Line 171 “silt”

Results: general…fine to report veg in tables or in supplemental, but specific discussions of species within text should only relate to measurements of production or correlations with soil variables?

In general…it’s hard to keep up with all the acronyms repeatedly discussed here. You might consider giving the grasslands their actual names.

Line 290. Do not repeat results in discussion.

Line 337 is called a tautology. Size fractions describe soil properties, because size fractions are soil properties.

Figure 1: shows nicely that TP, TN, and SOC are very autocorrelated. Can you include your BGB and AGB in this analysis?

Table 2 should be supplemental materials

Reviewer #2: The paper summarized study that compared plant and soil variables among several alpine grasslands in Tibet. The study surveyed 5 grassland types in August 2016. Alpine grasslands are important ecosystems worldwide and obviously vulnerable to climate change. Baseline information about how these grasslands respond to environmental changes will be important to know. Unfortunately, the study summarized here leaves much to be desired.

The paper itself is merely a descriptive comparison of grassland types and represents only one ‘snapshot’ of time – August 2016. While descriptive information about grassland ecosystems is not without value, it was hard for me to see the usefulness this particular information. For example, no information was given about the management history of these grasslands. Were they grazed, hayed, or fertilized? Or are they protected grasslands? If they are grazed, for how long, and at what stocking density? The lack of management information is a major omission as this would likely have a big impact on the site characteristics.

In addition to these issues, the data analysis was mostly inappropriate. What was the point of conducting ANOVAs across these grassland types? No treatments were applied to the grasslands so what were the authors trying to determine? In the context of a descriptive study like this, does it really matter that soil C was statistically different between several grassland types? Presenting the variable means and standard deviations for each would have sufficed. A data set like this is more amenable to a multi-variate approach, which the authors attempted with the RDA analysis. Again though, I fail to see how useful this analysis was given the short time frame and lack of management information provided.

The paper has other minor issues but given the problems mentioned above, I do not see the point in going through these. This data set might be publishable in a regional journal or other outlet, but I do not feel the paper, in its current form, has a great deal of value to an international audience. As a suggestion, the authors might consider beginning a long-term monitoring program with this data set and collect plant and soil data over many years. Showing how these variables might respond to climate variation or management activities over time would be valuable to the scientific community.

6. PLOS authors have the option to publish the peer review history of their article (what does this mean?). If published, this will include your full peer review and any attached files.

Reviewer #1: No

Reviewer #2: No

---

## [Author Response · Author response to Decision Letter 0]

25 Dec 2019

Dec 26, 2019

Prof. Tuneera Bhadauria

PLOS ONE

Dear Dr. Tuneera Bhadauria:

Re: Manuscript # PONE-D-19-25384 R1:

I have revised the manuscript based on editorial changes suggested by the reviewers. The revisions are track-changed in the revised manuscript. I hope that the revisions have adequately addressed the editor’s and reviewers’ concerns.

For field studies, no specific permissions were required for these locations and the field studies did not involve endangered or protected species. And the data was also uploaded in Supporting Information file.

Details of the revision conducted are in the following pages.

We appreciate very much your and the referees time and valuable suggestions and comments. We recognize that the comments have helped us to improve the quality of this manuscript. I look forward to your final decision on this manuscript for potential publication in the PLOS ONE. If you have any further comments or questions, please do not hesitate to contact me.

Yours sincerely

Beibei Zhang

Response to reviewers’ comments:

Reviewer: 1

Comments to the Author

This paper contains an interesting data set, but an uninteresting story. I have concerns about both plant and soil measurements, as the plant biomass seems low even for very arid systems, and the near absence of clay in the soils seems very unusual. The authors could be more convincing in discussing their methods, thereby providing some confidence in the numbers, and also by comparing their results to other, similar findings in the discussion.

- Accepted and Revised. We found some similar results about the lower clay fraction and used in the discussion.

Relating plant and soil characteristics can be strengthened. The correlation matrix among the soil variables should be included (at least as supplemental…). The authors should know that TP, TN and SOC are auto-correlated to the point where only one of the variables, along with C:N ratio, should be used in any regression relationship. Soil moisture, unless averaged over the seasons, is a “snapshot measurement” and tells us almost nothing about plant dynamics.

- Accepted and Revised. We added biomass data and to do the correlation with soil characteristics in Figure 3 which showed the correlation between plant growth and soil characteristics and we also deleted the soil water content in Figure 3 and Table 5.

The value of presenting findings of 11 communities within 5 grasslands is not discussed. If the variances in variables justifies this separation, then OK but this needs to be explicit.

- Accepted and Revised. We discussed this in the first part of Discussion. 

The literature cited on biodiversity-productivity relationships as well as on soil-plant feedbacks is dated. Newer summaries and syntheses are available. Species richness of these sites seems very, very low compared to other cold region/dry sites, and is worthy of comment. Historical legacies (i.e., impacts of vertebrate grazing) need to be presented so readers know of its significance (either large or small….).

- Accepted and Revised.

There could be a compelling story here by making comparisons to similar, published data on cold region herbaceous systems, but, alone, the work doesn't merit publication in an international journal. Sensitivity responses based upon measurements presented here are tenuous at best.

- Accepted and Revised.

Abstract: Line 38: The highest value (of what?) was always…

- Accepted and Revised.

Introduction, line 83: “influences stability” but does not “determine stability”.

- Accepted and Revised.

Lines 92-96. Actually, carbon sequestration can be independent, at least of AGB.

- Accepted and Revised.

Line 109: you have no data on change in soil texture so this statement doesn’t support your work.

- Accepted and Revised.

Line 171 “silt”

- Accepted and Revised.

Results: general…fine to report veg in tables or in supplemental, but specific discussions of species within text should only relate to measurements of production or correlations with soil variables? In general…it’s hard to keep up with all the acronyms repeatedly discussed here. You might consider giving the grasslands their actual names.

- Accepted and Revised. We changed the Table 2 as a supplemental. 

Line 290. Do not repeat results in discussion.

- Accepted and Revised.

Line 337 is called a tautology. Size fractions describe soil properties, because size fractions are soil properties.

- Accepted and Revised.

Figure 1: shows nicely that TP, TN, and SOC are very autocorrelated. Can you include your BGB and AGB in this analysis?

- Figure 1 didn’t have this information. Do you mean Figure 3? For Figure 3, we accepted and revised.

Table 2 should be supplemental materials

- Accepted and Revised.

Reviewer: 2

Comments to the Author

The paper summarized study that compared plant and soil variables among several alpine grasslands in Tibet. The study surveyed 5 grassland types in August 2016. Alpine grasslands are important ecosystems worldwide and obviously vulnerable to climate change. Baseline information about how these grasslands respond to environmental changes will be important to know. Unfortunately, the study summarized here leaves much to be desired.

- We agree with the reviewer’s comment. Because the funding and the other reason limited, we only got the data in 2016. If we have any other change we will propose the further research, with more seasonal and interannual investigations to evaluate the results.

The paper itself is merely a descriptive comparison of grassland types and represents only one ‘snapshot’ of time – August 2016. While descriptive information about grassland ecosystems is not without value, it was hard for me to see the usefulness this particular information. For example, no information was given about the management history of these grasslands. Were they grazed, hayed, or fertilized? Or are they protected grasslands? If they are grazed, for how long, and at what stocking density? The lack of management information is a major omission as this would likely have a big impact on the site characteristics.

- We agree with the reviewer’s comment. There is no special management for the grassland and which only grazed. I revised this in the Material and Methods part.

In addition to these issues, the data analysis was mostly inappropriate. What was the point of conducting ANOVAs across these grassland types? No treatments were applied to the grasslands so what were the authors trying to determine? In the context of a descriptive study like this, does it really matter that soil C was statistically different between several grassland types? Presenting the variable means and standard deviations for each would have sufficed. A data set like this is more amenable to a multi-variate approach, which the authors attempted with the RDA analysis. Again though, I fail to see how useful this analysis was given the short time frame and lack of management information provided.

- We agree with the reviewer’s comment. Because we want to see the differences between five grassland types under natural conditions, the ANOVA analysis was used to see their variance. For RDA analysis, although it sampled only one year it could reflect the relationship between soil properties and plant growth and it could be used as the basic data and reference compared to other research.

The paper has other minor issues but given the problems mentioned above, I do not see the point in going through these. This data set might be publishable in a regional journal or other outlet, but I do not feel the paper, in its current form, has a great deal of value to an international audience. As a suggestion, the authors might consider beginning a long-term monitoring program with this data set and collect plant and soil data over many years. Showing how these variables might respond to climate variation or management activities over time would be valuable to the scientific community.

- We agree with the reviewer’s comment. Since we only had one year’s data and we also want to show our data to other researcher, we compared our results to other similar findings in the discussion.

---

## [Editor Report · Decision Letter 1]

13 Jan 2020

Differences in species diversity, biomass, and soil properties of five types of alpine grasslands in the Northern Tibetan Plateau

PONE-D-19-25384R1

Dear Dr.beibei zhang

We are pleased to inform you that your manuscript has been judged scientifically suitable for publication and will be formally accepted for publication once it complies with all outstanding technical requirements.

With kind regards,

Tunira Bhadauria, Ph.D.

Academic Editor

PLOS ONE

Additional Editor Comments (optional):

I have gone through the revised manuscript of the authors and Even though the duration of the research carried out was of shorter time span but considering the geographical position of Tibet I think it will be interesting for international audience to know about the variations existing among grassland types and also between soil properties and species diversity there. The authors have revised the manuscript thoroughly and have incorporated all the comments and suggestions put forward by both the referees in the text and also in the figures and tables. I think the manuscript has sufficient merit to be accepted for publication in the Journal.
---

## [Editor Report · Acceptance letter]

29 Jan 2020

PONE-D-19-25384R1 

Differences in species diversity, biomass, and soil properties of five types of alpine grasslands in the Northern Tibetan Plateau 

Dear Dr. Zhang:

I am pleased to inform you that your manuscript has been deemed suitable for publication in PLOS ONE. Congratulations! Your manuscript is now with our production department. 

With kind regards,

on behalf of

Dr. Tunira Bhadauria 

Academic Editor

PLOS ONE